# Effects of Dietary Taurine Supplementation to Gilts during Late Gestation and Lactation on Offspring Growth and Oxidative Stress

**DOI:** 10.3390/ani9050220

**Published:** 2019-05-06

**Authors:** Mengmeng Xu, Long Che, Kaiguo Gao, Li Wang, Xuefen Yang, Xiaolu Wen, Zongyong Jiang, De Wu

**Affiliations:** 1Key Laboratory for Animal Disease-Resistant Nutrition of the Ministry of Education of China, Institute of Animal Nutrition, Sichuan Agricultural University, Chengdu 611130, Sichuan, China; xumengmeng2013@126.com (M.X.); chelong1989@126.com (L.C.); 2State Key Laboratory of Livestock and Poultry Breeding, Ministry of Agriculture Key Laboratory of Animal Nutrition and Feed Science in South China, Guangdong Public Laboratory of Animal Breeding and Nutrition, Guangdong Key Laboratory of Animal Breeding and Nutrition, Institute of Animal Science, Guangdong Academy of Agricultural Sciences, Guangdong 510640, Guangzhou, China; gaokaiguo312@126.com (K.G.); wangli1@gdaas.cn (L.W.); yangxuefen@gdaas.cn (X.Y.); wenxiaolu@gdaas.cn (X.W.)

**Keywords:** taurine, offspring, intestinal, barrier function, oxidative stress

## Abstract

**Simple Summary:**

Previous studies showed that gilts had elevated oxidative stress during late gestation and lactation, and could affect offspring growth. Taurine (Tau) is an important regulator of oxidative stress and possesses growth-enhancing properties. Our results suggested that taurine supplementation during late gestation and lactation of gilts increased growth performance in piglets through improved milk quality of gilts and intestinal morphology and barrier function of offspring.

**Abstract:**

Birth is one of the most important events of animal production agriculture, as newborns are abruptly forced to adapt to environmental and nutritional disruptions that can lead to oxidative damage and delay in growth. Taurine (Tau) is an important regulator of oxidative stress and possesses growth-enhancing properties. In the present study, we investigated the effects of dietary Tau supplementation in gilts during late gestation and lactation on the growth performance of piglets by assessing intestinal morphology and barrier function, and oxidative stress status. Sixteen gilts were randomly allocated to the Con (basal diet) and Tau (basal diet with 1% Tau) groups from 75 d of gestation to weaning. Maternal dietary Tau supplementation significantly increased weaning weight and average daily gain weight in piglets. Piglets in the Tau group had higher villus height and villus height-to-crypt depth ratio (VCR), ZO-1 protein expression, and secretory immunoglobulin A (sIgA) content in the jejunum. Meanwhile, Tau bebeficial affected the milk quality of gilts, as indicated by decreased malondialdehyde (MDA) concentration and increased total superoxide dismutase (T-SOD), total antioxidative capability (T-AOC), glutathione peroxidase (GPx), and catalase (CAT) activity. Furthermore, Tau supplementation increased T-SOD activity in plasma and SOD2 protein expression in the jejunum in the piglets. In conclusion, this study provides evidence that dietary Tau supplementation to gilts improves growth performance in piglets, owing to improved intestinal morphology and barrier function, as well as inhibition of oxidative stress.

## 1. Introduction

Birth is one of the most significant events in the life of mammals, as neonates are abruptly forced to adapt to environmental, immunological, and nutritional disruptions [1]. The birth transition of neonates is commonly accompanied by oxidative stress and impairment of the intestinal barrier [2]. Studies have also indicated that the nutritional value of important, e.g. immune globulin, in maintaining neonate health and growth [3], and maternal malnutrition can result in metabolic disorders [4], or disruptions in physiological oxidant–antioxidant equilibrium, thereby leading to oxidative stress [4], and eventually inducing growth retardation in offspring. Antibiotics can be effective in preventing and treating this problem, but their excessive use often results in the development of antibiotic resistance [5]. Hence, numerous alternatives to antibiotics have been investigated, among which functional amino acids have attracted considerable research interest because of their safety and specificity in mothers and neonates [6,7].

Taurine (Tau), a metabolite of methionine and cysteine [8], holds great potential for application in the food, pharmaceutical, and agricultural industries [9]. In particular, Tau has potential applications as a dietary supplement for animals [10]. Appreciation of the important role of Tau in regulating mammalian intestine development and growth rates has grown steadily in recent years [11,12]. For example, piglets fed Tau-supplemented diets show decreases intestinal permeability and apoptosis [13]. Unfortunately, the underlying mechanisms responsible for the improvements in offspring growth and intestine development owing to maternal Tau supplementation are not clearly elucidated. Tau also has numerous biological functions, including anti-inflammatory and anti-oxidative effects [14]. Previous studies have confirmed that Tau regulates oxidative stress by protecting against several environmental toxins and drug-induced multiple organ injuries [15,16]. These beneficial properties of Tau suggest that it may be an effective dietary additive, but its use as a maternal food supplement for humans and animals to improve offspring oxidative capacity is still not widespread or well understood. 

Accordingly, the present study was performed to explore the effects of gilt supplementation with dietary Tau on intestinal architecture, barrier function, and oxidative stress in piglets, aiming to provide evidence for the mechanisms by which Tau enhances growth performance in offspring. 

## 2. Materials and Methods

### 2.1. Animals and Diet

All experimental procedures followed the current laws regarding animal protection (Ethic Approval Code: 5YXK2016-0165) and were approved by the Guide for the Care and Use of Laboratory Animals prepared by the Animal Care and Use Committee of the Guangdong Academy of Agricultural Sciences. Sixteen Yorkshire × Landrace gilts, with a similar genetic background, back fat thickness (15.73 ± 0.75 mm) and bodyweight (149.88 ± 4.28 kg), were randomly allocated to control group (Con, basal diet with no added Tau) or Tau-supplemented diet group (Tau, basal diet + 1% Tau, provided by the Yongan Pharmaceutical Co., Ltd., Wuhan, China) from 75 d of gestation until weaning (at 21 d of lactation). The basal diet during late gestation and lactation were formulated to meet nutrient requirements as recommended by the National Research Council 2012 (NRC 2012. During late gestation: Digestible energy, 13.05 MJ/kg; crude protein, 13.60%; Ca, 0.92%; P, 0.38%; during lactation: digestible energy, 13.89 MJ/kg; crude protein, 18.54%; Ca, 1.01%; P, 0.47%). In addition, nitrogen balance was achieved by adding alanine. Gilts were housed in individual gestation stalls from 75 until 110 d of gestation. At 110 d of gestation, gilts were moved to farrowing crates that had a piglet creep area provided with heat lamp and nipple water. Gilts were fed an average of 2.72 kg/d before farrowing. For lactating gilts, starting from the day after farrowing, a lactation diet was provided 3 times daily and restricted with the amount increasing gradually until 5 d of lactation, whereupon it was provided ad libitum. Water was available ad libitum throughout. Within 24 h of farrowing, litter size was standardized to 11 piglets per sow, depending on their weight and availability, by cross-fostering within the same treatment group. When piglet mortality happened, the dead piglets would be replaced by another piglet with similar bodyweight and age. 

### 2.2. Measurement of Body Weight, Back Fat Thickness, and Feed Intake

Gilts were weighed, and P_2_ back fat thickness (6.5 cm from the midline over the last rib) was measured at 1 and 21 d of lactation using A-mode ultrasonography (Lean-Meater; Renco Corporation, Minneapolis, MN, USA). At farrowing, the numbers of births, the number of live births, and neonatal weight (before colostrum consumption) were recorded. The daily feed intake of the gilts was recorded. 

### 2.3. Collection of Blood and Milk 

Blood samples of gilts were collected via ear venipuncture in the morning before feeding at 1 and 10 d of lactation from eight gilts per treatment. In addition, blood samples were collected from new born piglets (*n* = 8 per treatment, 1 piglet per litter and weight close to the average level of per treatment), 7 d post-birth piglets (*n* = 8 per treatment, 1 piglet per litter and weight close to the average level of per treatment), and 7 d post-birth piglets (*n* = 8 per treatment, 1 piglet per litter and weight close to the average level of per treatment) by jugular venipuncture. The new born piglets were removed from their mothers immediately after birth without sucking colostrum for sample collection. All blood samples were centrifuged immediately after collection (3000× *g* for 15 min at 4 °C). Plasma samples were collected and stored at −20 °C for future analysis. Milk was collected and pooled from all the functional glands on the left side after thoroughly cleaning the udder. After parturition, colostrum was collected by hand-milking before any piglet suckled the mammary teats. During lactation, milk samples were collected before feeding in the morning at 10 d of lactation after gilts were given 10 IU oxytocin (Sansheng Biological Technology, Ningbo, Zhejiang, China) via ear venipuncture. 

### 2.4. Tissue Sample Collection

Typical piglets with weight close to the average level of per treatment were selected from each gilt for blood sample collection and slaughter at birth, 7 post-birth and 21 d post-birth (n = 8). These piglets were anesthetized with an intravenous injection of Zoletil 50 at a dose of 0.1 mg/kg body weight and slaughtered. Duodenal, jejunum, and ileum samples with a length of approximately 2 cm were cut and fixed in a 4% paraformaldehyde solution for histological analyses. The rest of the jejunum, ileum, and duodenum was stored in a cryopreservation tube, frozen in liquid nitrogen, and stored at −80 °C for further analysis.

Four percent paraformaldehyde-fixed duodenal, jejunum, and ileum samples were dehydrated and embedded in paraffin. Cross-sections of each sample were prepared, stained with hematoxylin, hydrochloric acid, and eosin, and sealed with neutral resin. Villus height and crypt depth were examined in 4 μM of the duodenal, jejunum and ileum samples using an image processing and analysis system (Image-Pro, Media Cybernetics, Inc., Bethesda, MD, USA). The length from the tip of the villus to the villus-crypt junction was classified as villus height, and the invaginated depth between adjacent villi was classified as crypt depth. Ten intact, well-oriented, crypt-villus units were analyzed per intestinal segment. The values obtained from 10 villi in each intestinal segment were averaged. The villus height-to-crypt depth ratio (VCR) was computed from the measurements obtained above.

### 2.5. Oxidative Stress Biomarker Analyses

Malondialdehyde (MDA), total anti-oxidative capability (T-AOC), total superoxide dismutase (T-SOD), glutathione peroxidase (GPx), and catalase (CAT) content in all milk and plasma samples was determined using commercial kits (Jiancheng Institute of Biological Technology, Nanjing, Jiangsu, China) according to the manufacturer’s instruction. The jejunum samples of piglets were crushed with a mortar and pestle in liquid nitrogen, followed by the commercial kits analyses (Jiancheng Institute of Biological Technology, Nanjing, Jiangsu, China).

### 2.6. Gene Expression Analyses

RNA was isolated using TRIzol reagent (Invitrogen, Carlsbad, CA, USA) and RNeasy Mini kit (RR037A; Takara Biotechnology, Kusatsu, Japan) according to the manufacturer’s instruction. RNA was reverse transcribed to cDNA with reagents obtained from TaKaRa Biotechnology. The mRNA levels were analyzed with a 7900HT Fast Real-Time PCR system (Thermo Fisher Scientific, Waltham, MA, USA) using SYBR Green Real-Time PCR reagent (RR820A; Takara Biotechnology, Dalian, Liaoning, China). Gene expression levels were normalized to β-actin expression levels. The cycle threshold (2^−△△Ct^) method was used to calculate the relative gene expression. The sequences of the primers were given in Table 1.

### 2.7. Western Blot

To obtain total protein lysates, the cells were lysed using RIPA buffer (Beyotime, Shanghai, China) on ice for 30 min. All lysates were collected with cell scrapers and centrifuged at 4 °C at 12,000× *g* for 10 min to remove any debris. Protein concentrations were measured using BCA protein assay kit (Thermo Fisher Scientific, Waltham, MA, USA). Proteins (50 μg) were mixed with Laemmli sample buffer and boiled at 100 °C for 10 min. The proteins were separated by electrophoresis using 10% sodium dodecyl sulfate-polyacrylamide gel electrophoresis (SDS-PAGE) at 70 V for 40 min or 110 V for 60 min and transferred onto a polyvinylidene difluoride (PVDF) membranes at 250 mA for 90 min. The membranes were blocked in QuickBlockTM Western (Beyotime, Shanghai, China) for 1 h at 25 °C. Membranes were incubated overnight at 4 °C with the primary antibodies. The membranes were washed 3 times with Tris-buffered saline for 10 min and incubated for 1 h at room temperature with appropriate HRP-conjugated reporter antibodies. After three 10-min washes, VersaDoc imaging system (Bio-Rad, Hercules, CA, USA) was used to visualize immunoreactivity with a chemiluminescent HRP substrate (Millipore, Billerica, MA, USA). The band intensities were determined using ImageJ software and expressed relative to β-actin. All antibodies were purchased from Abcam (Cambridge, MA, USA).

### 2.8. ELISA Analysis of Secretory Immunoglobulin A *(sIgA)*

The amount of sIgA was determined using ELISA kits in accordance with the manufacturer’s instructions (Jiancheng Institute of Biological Technology, Nanjing, Jiangsu, China).

### 2.9. Statistical Analyses 

All experimental data were presented as means ± SEM and analyzed using the SPSS statistical software program (v. 19.0 for windows, SPSS; IBM SPSS Company, Chicago, IL, USA). Individual gilts or a litter of piglets were considered the experimental unit. A general linear model with litter size at birth was used as the covariate for average birth weight, and litter size at weaning was used as the covariate for the average weaning weight of remaining piglets. Other data were analyzed by an independent-sample *t*-test. The level of significance was set at *p* < 0.05.

## 3. Results

### 3.1. Gilt and Litter Growth Performance 

As shown in Table 2, although Tau supplementation did not have a significant effect (*p* > 0.05) on back fat thickness and weight, but the average daily feed intake was higher (*p* < 0.05) in the supplemented gilts during the lactation period. 

While Tau supplementation did not have a significant effect (*p* > 0.05) on total number of births, number of live births, or neonatal weight, the average daily gain (*p* < 0.01) during lactation and weight (*p* < 0.05) increased at 7 and 21 d of lactation of piglets (Table 2). 

### 3.2. Enzyme Activities in Plasma and Milk of Gilts

Dietary Tau supplementation led to significantly lower (*p* < 0.05) MDA content in plasma at 1 and 10 d of lactation than that in the Con group, however, T-SOD, T-AOC, GPx, and CAT activities in plasma were higher (*p* < 0.01) at 10 d of lactation, but no difference (*p* > 0.05) was noted in the plasma antioxidant index at 1 d of lactation in gilts (Table 3). The MDA content in milk was significantly lower (*p* < 0.05) in the Tau group at 1 and 10 d of lactation in gilts than in the Con group, however, T-SOD, T-AOC, GPx, and CAT activity in milk were higher (*p* < 0.01) at a and 10 d of lactation (Table 3).

### 3.3. Intestinal Morphological Observation of Piglets 

The villus height was significantly higher (*p* < 0.05) at the duodenum, jejunum, and ileum segment in the Tau group at 1 and 21 d post-birth in piglets than in the Con group, but no significant effects (*p* > 0.05) on villus height of duodenum and ileum were detected at 7 d post-birth (Table 4 and Figure 1). Crypt depth did not differ significantly (*p* > 0.05) between the two groups, except in the duodenum at 7 d post-birth in piglets. Maternal dietary Tau supplementation resulted in a significantly higher (*p* < 0.05) VCR at the duodenum, jejunum, and ileum at 21 d post-birth than that in the Con group. 

### 3.4. Enzyme Activities in Plasma of Piglets

For gilts supplemented with Tau, piglets had higher (*p* < 0.01) T-SOD, T-AOC, GPx, and CAT activity and lower (*p* < 0.05) MDA content at 1 d post-birth (Table 5). In addition, piglets in the Tau group had higher T-SOD and T-AOC activity (*p* < 0.05) at 7 d post-birth than did the Con group. Moreover, gilts supplemented with Tau had higher (*p* < 0.05) plasma activity of T-SOD and lower (*p* < 0.05) MDA content at 21 d post-birth (Table 5). 

### 3.5. Antioxidant Indicators in the Jejunum

The MDA content of jejunum was markedly lower (*p* < 0.01), while T-SOD activity was higher (*p* < 0.05) in the Tau group at 1, 7, and 21 d post-birth than in the Con group. Maternal Tau supplementation caused T-AOC activity to increase (*p* < 0.01) on 1 and 7 d post-birth, but no difference (*p* > 0.05) was noted 21 d post-birth. In addition, there was no difference (*p* > 0.05) in GPx activity between the two treatments (Table 6). 

The expression of oxidative stress status-related genes in the jejunum of the piglets was altered at all the time points. As shown in Figure 2A, maternal supplementation with Tau caused higher (*p* < 0.05) SOD2 mRNA expression at three time points, but no effects were detected on the SOD1 and GPx mRNA in piglets at 1 and 7 d post-birth. However, Tau group had significantly higher GPx mRNA expression 21 d post-birth (Figure 2A). The effects of Tau supplementation on jejunum antioxidant protein expression is shown in Figure 2B. Tau supplementation led to higher SOD2 protein abundance than that in the Con group, consistent with the results for gene expression.

### 3.6. ZO-1 Protein Expression and sIgA Production in the Jejunum

Tau supplementation led to higher ZO-1 protein abundance at 21 d post-birth than that in the Con group, but not at 1 and 7 d post-birth (Figure 3A). As shown in Figure 3B, maternal dietary Tau supplementation failed to enhance jejunum sIgA production at 1 and 7 d post-birth, but it enhanced the jejunum sIgA production at 21 d post-birth.

## 4. Discussion

A few studies have suggested that maternal dietary intake can regulate intestinal outcomes in offspring [17,18], at least in part via changes to intestinal oxidative status and barrier function [19]. While an appropriate balance of free radicals is thought to be necessary for optimal development, excess free radicals are generally considered to be harmful [20]. Intestinal structure, function, and metabolism can change from birth to weaning in both humans and animals depending on maternal dietary intake and milk quality [21]; therefore, increased oxidative stress in milk via maternal dietary intake leads to metabolic absorption disorders in adults [22]. In order to explore the molecular mechanisms mediating the effect of maternal dietary manipulations on intestinal health in the offspring, the effect of nutrient intake, including oxidative stress, intestinal morphology, and barrier was examined.

Weaning weight is a key indicator of the reproductive performance of gilts, and is important in controlling their economic efficiency. Strong evidence from animal studies indicates that maternal manipulations could result in changes in the growth rates of offspring [23]. In the current study, a greater weaning weight was noticed after Tau supplementation, accompanied by an increased average daily weight gain, indicating that maternal Tau supplementation improved the growth rate of offspring. Here, we also noted that the average daily feed intake of gilts was higher in the Tau group than in the Con group, suggesting that Tau may have the function of promoting food intake. Moreover, to induce growth in offspring, mothers usually secrete more milk [24]. While milk yield was not determined in this study, intake, loss of back fat, and weight loss in gilts could reflect milk yield to a certain extent, as the nutrition in sows is divided between milk constituent synthesis [25] and self-body weight gain. Our results suggest that maternal dietary supplementation with Tau could improve the weaning weight of offspring via increased maternal milk yield. 

Maternal milk quality is an important factor limiting neonate growth up to weaning [26]. However, the majority of experiments to date have revealed that oxidative stress can damage normal milk function via changes in certain substances [27], including decreased immunoglobulin content. Therefore, protecting normal milk functions from oxidative stress is indispensable in the maintenance of neonate development [28]. Accordingly, we investigated whether maternal Tau diets could mitigate oxidative damage in milk by evaluating antioxidant-related parameters. Dietary Tau supplementation significantly increased the activity of representative enzymatic antioxidant activities (T-SOD, T-AOC, GPx, and CAT) and reduced the levels of oxidative damage markers (MDA) in milk at 1 and 10 d of lactation, suggesting that Tau ingestion could partly enhance the function of antioxidant defense systems by improving enzymatic antioxidant activity. Milk nutrition during lactation is closely associated with neonate growth and development in humans and animals, because it alters neonate metabolism and health [29,30]. We next tested antioxidant capacity in neonates to further understand how maternal Tau supplementation could affect offspring development. Tau increased the antioxidant capacity from birth to weaning in plasma. Interestingly, MDA content of plasma was reduced in both treatment groups from birth to weaning of piglets. The reason may be that piglets gradually adapt to the environment and piglets’ intestinal development was more mature in weaning compared with the newborn stage. The formation of free radicals represents a form of physiological adaptation to the environment, with important functions as second messengers in controlling superoxide and nitric oxide function [31]. However, excessive free radical formation or damage to the antioxidant system result in severe intestinal pathology [32]. Oxidative stress can increase intestinal damage and destroy intestinal health; however, the antioxidant system helps regulate and control the levels of free radicals at the required physiological concentrations [33]. The present study showed that maternal Tau ingestion increased the antioxidant factor SOD2 mRNA and protein abundances from birth to weaning and increased GPx mRNA expression at weaning. Thus, Tau inhibition of intestinal oxidative damage may increase antioxidant factors. Our findings elucidate the positive role of Tau in conferring resistance to oxidative stress in both mother and offspring.

Undesirable alterations in intestinal architecture such as villus atrophy and crypt hyperplasia are commonly encountered as a result of oxidative stress [34]. A decrease in the VCR or villus height is considered deleterious for digestion and absorption and can lead to retarded growth in neonates [35]. Consequently, maintaining normal intestinal architecture and function is essential for growth and development in neonates. Thus, increased VCR in the duodenum, jejunum, and ileum at weaning as well as increased villus height was detected in the offspring of mothers fed Tau-supplemented diets. These observations support the notion that maternal Tau supplementation can change intestinal morphological structure and promote intestinal digestion-absorption function in offspring. Adding 0.3% Tau to the diet of piglets resulted in significantly improved intestinal function, corroborating the before mentioned view [13]. Those findings suggested that the growth-promoting effects of Tau on offspring can be partially attributable to the improved intestinal morphology and function. 

There is extensive evidence that oxidative stress is correlated with impaired intestinal barrier function in neonates [36,37]. Interestingly, maternal dietary supplementation with Tau provides a promising approach to improve intestinal barrier function in offspring. Therefore, we expected that Tau would have benefits on intestinal barrier function of offspring when administered to mothers. ZO-1 is a specific protein found in the small intestine, involved in regulation and maintenance of tight junction structure [38], contributing to the selective permeability of the barrier, conferring physical barrier function to the intestine [39]. In the present study, higher ZO-1 protein expression in the jejunum at weaning was noticed after maternal Tau supplementation, suggesting that maternal dietary supplementation with Tau could enhance the intestinal physical barrier function of offspring. Tau supplementation increased jejunum sIgA content at weaning, suggesting that maternal dietary supplementation with Tau could enhance the intestinal immune barrier function of offspring [40,41]. These results revealed that Tau contributes to repair the oxidative stress-associated intestinal barrier dysfunction in piglets and possibly to improve their growth performance.

## 5. Conclusions

The present study showed that supplementing the maternal diet with 1% Tau in gilts improved the antioxidant ability, intestinal morphology, and barrier function of offspring. Furthermore, these changes were accompanied by an enhanced growth performance in offspring. Our observations provide a convincing scientific basis for Tau acting as a functional amino acid and affecting growth promoters in pig production.

## Figures and Tables

**Figure 1 animals-09-00220-f001:**
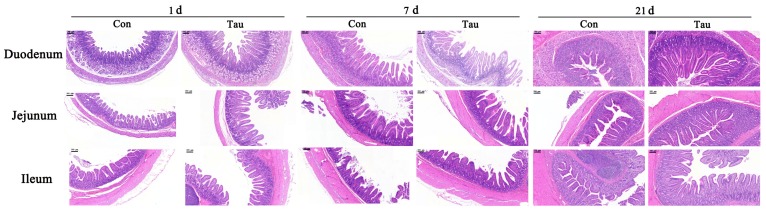
Effect of dietary Tau supplementation during late gestation and lactation on the intestinal histomorphology of piglets. Con, basal diets during gestation and lactation; Tau, basal diets supplemented with 1% Tau during gestation and lactation. The original magnification is 50 x and scale bar is 200 μm.

**Figure 2 animals-09-00220-f002:**
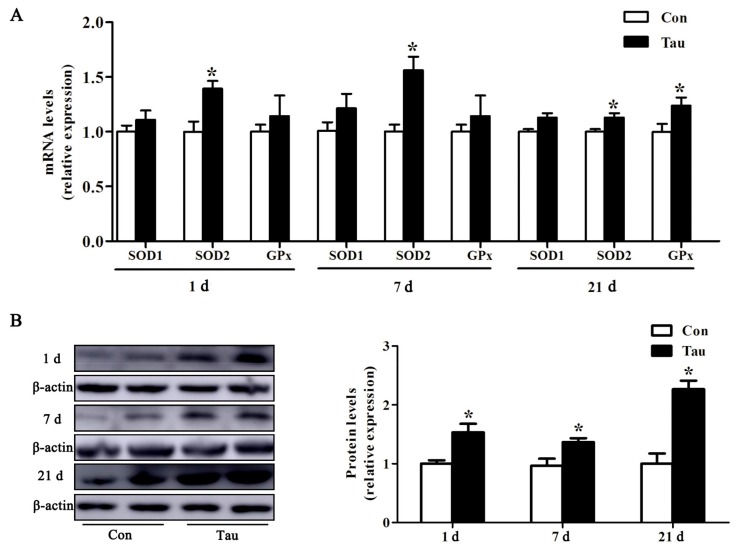
Relative antioxidant indicators expressions in jejunum of piglets. (**A**) mRNA expression; (**B**) SOD2 protein expression. Con, basal diets during gestation and lactation; Tau, basal diets supplemented with 1% taurine during gestation and lactation. SOD1, superoxide dismutase D1; SOD2, superoxide dismutase D2; and GPx, glutathione peroxidase.

**Figure 3 animals-09-00220-f003:**
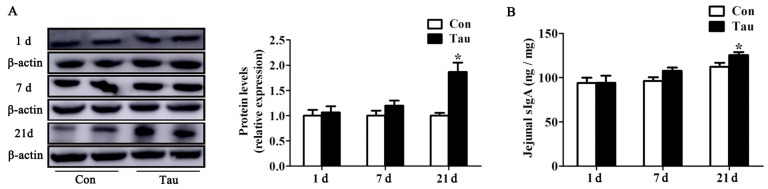
Effect of jejunum barrier function after maternal taurine supplementation in piglets. (**A**) ZO-1 protein expression; (**B**) sIgA contents was analyzed by ELISA kits. Con, basal diets during of gestation and lactation; Tau, basal diets supplemented with 1% taurine during of gestation and lactation.

**Table 1 animals-09-00220-t001:** Primer sequences of the target and reference genes.

Gene	Primers	Sequence	Accession Number	Product Size (bp)
SOD1	Forward	5’-GAGACCTGGGCAATGTGACT-3’	GU944822.1	139
	Reverse	5’-AGCGACGCTACGTTCTCAAT-3’		
SOD2	Forward	5’-TGGAGGCCACATCAATCATA-3’	NM_214127.2	136
	Reverse	5’-AGCGGTCAACTTCTCCTTGA-3’		
GPx	Forward	5’-GCTCGGTGTATGCCTTCTCT-3’	NM_214201.1	103
	Reverse	5’-AGCGACGCTACGTTCTCAAT-3’		
β-actin	Forward	5’-GGCCGCACCACTGGCATTGTCAT-3’	DQ845171.1	104
	Reverse	5’-AGGTCCAGACGCAGGATGGCG-3’		

**Table 2 animals-09-00220-t002:** Effects of dietary Tau supplementation during late gestation and lactation on the performance of gilts and piglets.

Items	Con	Tau	*p*-value
Gilts
Average daily feed intake in lactation, kg/d	4.84 ± 0.11	5.23 ± 0.10	0.015
Backfat thickness loss in lactation, mm	2.63 ± 0.82	2.88 ± 0.30	0.781
Weight loss in lactation, kg	8.69 ± 2.47	6.31 ± 0.65	0.380
Litter size
Total born	11.13 ± 0.55	11.37 ± 0.42	0.723
Alive at birth	10.88 ± 0.40	11.25 ± 0.37	0.540
Piglets
Neonatal weight, kg	1.26 ± 0.09	1.46 ± 0.06	0.099
Weight at birth after adjust, kg	1.28 ± 0.07	1.30 ± 0.07	0.899
Weight at day 7 of post-birth	2.56 ± 0.15	3.01 ± 0.09	0.031
Average daily gain, g	194.62 ± 5.92	230.11 ± 4.18	0.001
Weaning weight, kg	5.35 ± 0.14	6.29 ± 0.07	0.001

Con, basal diets during gestation and lactation; Tau, basal diets supplemented with 1% taurine during gestation and lactation.

**Table 3 animals-09-00220-t003:** Effect of dietary Tau supplementation during late gestation and lactation in plasma and milk enzyme activities of gilts at 1 and 10 d of lactation.

Items	1 d of Lactation		*p*-value	10 d of Lactation		*p*-value
	Con	Tau		Con	Tau	
Plasma						
MDA,nmol/ml	4.33 ± 0.61	2.77 ± 0.20	0.028	3.73 ± 0.15	3.05 ± 0.17	0.013
T-SOD,U/ml	75.31 ± 12.58	84.64 ± 11.20	0.585	80.57 ± 11.79	134.58 ± 14.17	0.008
T-AOC,U/ml	94.09 ± 10.80	79.83 ± 12.23	0.391	106.21 ± 14.98	165.16 ± 14.52	0.010
GPx,U	85.35 ± 9.00	80.21 ± 10.57	0.714	69.56 ± 8.08	125.51 ± 12.86	0.002
CAT,U/ml	0.56 ± 0.04	0.57 ± 0.04	0.183	0.50 ± 0.02	0.55 ± 0.01	0.003
Milk						
MDA,nmol/ml	4.48 ± 0.28	3.64 ± 0.69	0.026	16.15 ± 1.92	10.27 ± 1.36	0.031
T-SOD,U/ml	95.51 ± 13.83	161.84 ± 18.21	0.009	23.71 ± 0.67	29.48 ± 8.45	<0.001
T-AOC,U/ml	106.21 ± 14.98	165.16 ± 14.52	0.010	34.45 ± 0.90	105.93 ± 11.01	<0.001
GPx,U	103.75 ± 13.69	174.03 ± 17.87	0.005	38.32 ± 0.58	100.78 ± 8.11	<0.001
CAT,U/ml	0.69 ± 0.05	0.74 ± 0.08	0.016	0.64 ± 0.00	0.68 ± 0.03	<0.001

Con, basal diets during gestation and lactation; Tau, basal diets supplemented with 1% taurine during gestation and lactation. MDA, determine malondialdehyde; T-SOD, total superoxide dismutase; T-AOC, total antioxidative capability; GPx, glutathione peroxidase; and CAT, catalase.

**Table 4 animals-09-00220-t004:** Effect of dietary Tau supplementation during late gestation and lactation on the intestinal morphology of piglets.

Items	1 d of Post-Birth		*p*-value	7 d of Post-Birth		*p*-value	21 d of Post-Birth		*p*-value
	Con	Tau		Con	Tau		Con	Tau	
Villous height (µm)									
Duodenum	249.10 ± 24.07	503.08 ± 12.29	<0.001	584.95 ± 21.66	639.18 ± 21.96	0.101	403.74 ± 16.32	828.16 ± 34.66	<0.001
Jejunum	238.36 ± 9.66	284.00 ± 15.19	0.048	358.39 ± 12.47	524.04 ± 25.65	<0.001	469.16 ± 47.62	650.26 ± 28.57	0.007
Ileum	327.43 ± 9.60	397.12 ± 13.13	0.001	431.11 ± 29.72	453.31 ± 33.23	0.626	318.61 ± 12.22	467.21 ± 23.24	<0.001
Crypt depth (µm)									
Duodenum	176.48 ± 10.08	180.74 ± 8.27	0.749	256.69 ± 16.86	197.45 ± 17.45	0.029	263.05 ± 15.78	241.80 ± 34.76	0.590
Jejunum	140.55 ± 8.09	163.08 ± 11.33	0.128	152.38 ± 12.31	133.00 ± 9.26	0.231	267.34 ± 38.45	180.46 ± 21.93	0.075
Ileum	124.55 ± 4.72	121.69 ± 4.07	0.653	198.62 ± 6.88	198.06 ± 10.54	0.965	263.05 ± 15.78	266.80 ± 16.40	0.871
VCR									
Duodenum	1.47 ± 0.21	2.81 ± 0.12	<0.001	2.34 ± 0.16	3.55 ± 0.54	0.061	1.08 ± 0.07	2.39 ± 0.26	0.001
Jejunum	1.81 ± 0.11	1.77 ± 0.10	0.818	1.38 ± 0.06	1.99 ± 0.08	<0.001	2.33 ± 0.43	3.40 ± 0.29	0.049
Ileum	2.66 ± 0.15	3.29 ± 0.17	0.014	2.18 ± 0.14	2.35 ± 0.23	0.529	1.25 ± 0.10	1.79 ± 0.13	0.005

Con, basal diets during gestation and lactation; Tau, basal diets supplemented with 1% taurine during gestation and lactation; VCR, villus height-to-crypt depth ratio.

**Table 5 animals-09-00220-t005:** Effect of dietary Tau supplementation during late gestation and lactation on plasma enzyme activities of piglets.

Items	1 d of post-birth		*p*-value	7 d of post-birth		*p*-value	21 d of post-birth		*p*-value
	Con	Tau		Con	Tau		Con	Tau	
MDA,nmol/ml	6.02 ± 0.77	4.06 ± 0.13	0.032	6.38 ± 0.50	5.50 ± 1.19	0.135	6.47 ± 0.39	5.32 ± 0.30	0.042
T-SOD,U/ml	35.53 ± 4.78	104.92 ± 10.94	<0.001	33.43 ± 1.03	40.09 ± 2.38	0.017	28.32 ± 1.37	33.03 ± 1.17	0.016
T-AOC,U/ml	23.45 ± 0.50	41.22 ± 4.05	<0.001	34.45 ± 0.90	105.93 ± 11.01	<0.001	79.83 ± 12.23	94.09 ± 10.80	0.391
GPx,U	29.46 ± 1.16	39.29 ± 2.54	0.002	36.65 ± 0.90	36.65 ± 0.72	0.267	34.86 ± 0.73	36.91 ± 0.80	0.071
CAT,U/ml	0.34 ± 0.00	0.38 ± 0.00	<0.001	0.44 ± 0.00	0.42 ± 0.00	0.352	0.33 ± 0.00	0.34 ± 0.00	0.340

Con, basal diets during gestation and lactation; Tau, basal diets supplemented with 1% taurine during gestation and lactation. MDA, determine malondialdehyde; T-SOD, total superoxide dismutase; T-AOC, total antioxidative capability GPx, glutathione peroxidase; and CAT, catalase.

**Table 6 animals-09-00220-t006:** Effect of dietary taurine supplementation during late gestation and lactation on antioxidant indicators in the jejunum of piglets.

Item	1 d of Post-Birth		*p*-value	7 d of Post-Birth		*p*-value	21 d of Post-Birth		*p*-value
	Con	Tau		Con	Tau		Con	Tau	
MDA,nmol/mg	8.86 ± 0.31	7.08 ± 0.38	0.003	7.48 ± 0.20	6.04 ± 0.29	0.001	5.04 ± 0.37	3.39 ± 0.19	0.002
T-SOD,U/mg	96.88 ± 3.67	119.88 ± 2.98	<0.001	80.63 ± 3.05	93.25 ± 4.77	0.046	87.84 ± 5.36	108.54 ± 3.84	0.008
T-AOC,U/mg	1.23 ± 0.11	1.96 ± 0.11	<0.001	1.29 ± 0.06	1.86 ± 0.13	0.001	1.09 ± 0.10	1.26 ± 0.07	0.185
GPx,U	2.74 ± 0.15	2.86 ± 0.21	0.631	5.61 ± 0.38	6.54 ± 0.63	0.147	3.69 ± 0.25	4.63 ± 0.41	0.076

Con, basal diets during gestation and lactation; Tau, basal diets supplemented with 1% taurine duringgestation and lactation. MDA, determine malondialdehyde; T-SOD, total superoxide dismutase; T-AOC, total antioxidative capability; and GPx, glutathione peroxidase.

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
