# Peer review of "Effects of Dietary Taurine Supplementation to Gilts during Late Gestation and Lactation on Offspring Growth and Oxidative Stress"

_animals, 2019, doi:10.3390/ani9050220_

Round 1

Reviewer 1 Report

This is an interesting and important study that uses pigs as a model for parallel findings of maternal and offspring growth and vigor for humans and other species. The methodology is sound and sophisticated assays of oxidative stress biomarkers, gene expression, Western Blot, sIgA,  and intestinal histopathology are employed.

Specific suggestions/questions for revision

Lines 18-19. Suggest  " --Tau is an important regulator of oxidative stress and --"

Lines 23-25.  Suggest " ---most important events of animal production agriculture, as newborns are abruptly forced to adapt to environmental and nutritional disruptions that can lead to oxidative damage and delay in growth.  Taurine (Tau) is an important regulator of oxidative stress and --".

Line 28. "-- morphology and barrier function -- "

Line 33. "-- in the jejunum. However, Tau adversely --- "

Line 51." -- excessive use often results --"

Line 64. Delete "is"  Line 68. Delete "Therefore, further exploration is necessary."  

Line 71.  Delete  " -- partial theoretical --"  Lines 72-73. Delete this  sentence.   Lines 94-95. Good protocol. 

Line 108. Curious -- why the left side?  

Line 113. Please clarify what is meant here.

Line 134. Suggest "-- nitrogen, followed by the -- "

Line 168.  Suggest " The amount of sIgA -- "

Line 265. Suggest  " A few studies have suggested --" 

Line 275-276. Suggest  " --gilts, and is important in controlling their economic efficiency." 

Line 290. What "certain substances"?  

Line 308.  Word missing here. Line 310. Delete "be"

Line 323. Suggest  "Those findings suggested --- " 

Line 337. Suggest " -- to repair the -- "  Line 338. Suggest "-- improve their growth -- "  Line 343. Suggest "convincing " rather than "strong" 

Author Response

Reviewer #

1. Lines 18-19. Suggest  " --Tau is an important regulator of oxidative stress and --"

RESPONSE: Thank you very much for this comment. I have corrected it (lines 17-18).

2. Lines 23-25.  Suggest " ---most important events of animal production agriculture, as newborns are abruptly forced to adapt to environmental and nutritional disruptions that can lead to oxidative damage and delay in growth.  Taurine (Tau) is an important regulator of oxidative stress and --".

RESPONSE: Thank you for your suggestions. I have corrected it (lines 22-25).

3. Line 28. "-- morphology and barrier function -- "

RESPONSE: Thank you for your suggestions. I have corrected it  (line 27).

4. Line 33. "-- in the jejunum. However, Tau adversely --- "

RESPONSE: Thank you for your suggestions. I have corrected it (line 33).

5. Line 51." -- excessive use often results --"

RESPONSE: Thank you for your suggestions. I have corrected it (line 52).

6. Line 64. Delete "is"  Line 68. Delete "Therefore, further exploration is necessary."  

RESPONSE: I agree with your point, so I have deleted it.

7. Line 71.  Delete  " -- partial theoretical --"  Lines 72-73. Delete this  sentence.   Lines 94-95. Good protocol. 

RESPONSE: I agree with your point, so I have deleted it.

8. Line 108. Curious -- why the left side?  

RESPONSE: Both the left and right are feasible. The experiment just randomly selected one side. The oxytocin action time is short. If the milk was collected and pooled from all the functional glands in both side, the time is not enough. Therefore, in order to ensure the same sampling conditions, the experiment selects the left side.

9. Line 113. Please clarify what is meant here.

RESPONSE: We have changed “At 1, 7, 21 D post-birth, selected an average weight of piglets slaughter in each group at each time point” to “Typical piglets with weight close to the average level were selected from each sow for sacrifice at birth, 7 post-birth and 21 D post-birth (n = 8) (lines 120-121).

10. Line 134. Suggest "-- nitrogen, followed by the -- "

RESPONSE: I have corrected it (line 142).

11. Line 168.  Suggest " The amount of sIgA -- "

RESPONSE: I have corrected it (line 176).

12. Line 265. Suggest  " A few studies have suggested --" 

RESPONSE: I have corrected it (line 275).

13. Line 275-276. Suggest  " --gilts, and is important in controlling their economic efficiency." 

RESPONSE: I have corrected it (lines 285-286).

14. Line 290. What "certain substances"?  

RESPONSE: We are sorry about that we did not provide a clear description. I have added related content (lines 300-301).

15. Line 308.  Word missing here. Line 310. Delete "be"

RESPONSE: Thanks for your reminder. I have corrected it (line 321).

16. Line 323. Suggest  "Those findings suggested --- " 

RESPONSE: I agree with your point, so I have correct it (line 336).

17. Line 337. Suggest " -- to repair the -- "  Line 338. Suggest "-- improve their growth -- "  Line 343. Suggest "convincing " rather than "strong" 

RESPONSE: Thank you for your suggestions. We have already revised (lines 350-356).

Reviewer 2 Report

Line 33 – are you sure about the word “adversely”?

Line 47. The sentence started in this line is not clear probably because of missing words (after “important”?). Please rephrase.

Line 80: Gilts are female pigs in their first gestation. Those were the animals used in the trial? If so, why do you say “average parity”? The average body weight should be given.

Line 83: replace “to the end of weaning” by “until weaning”

Line 95: please explain better the criteria of cross-fostering, when you say “depending on their weight” your goal was?

Lines 103-111: Several explanations are needed:

a) when you say “select average weight per litter” that means that it was a piglet with a body weight close to the average of that litter?

b) if so there were differences between selected piglets? (e.g. in a litter A the mean was 1.1kg while in litter B the mean was 1.4kg – this is related also with explanation in line 95);

c)the blood was always obtained from the same piglets of each litter? If so couldn’t (shouldn’t) you perform repeated measures analyses of data?

d) the piglet blood samples collected on D1 were obtained before or after colostrum intake? If after, it could influence the enzymes results?

e) how were maintained the piglets after birth and before they could get access to colostrum?

f) the oxytocin administrated should be indicated in IU and the commercial product indicated.

Line 113: again clarify piglet selection. It was also important to indicate in the results section the weights (and variability) of slaughtered piglets at each age.

Line 134-135: please rephrase or complete the sentence.

Line 181: please rephrase. Suggestion: on back fat thickness and weight, but the average daily feed intake was higher (p < 0.05) in the supplemented gilts during period lactation

Table 1: I consider important to have in this table: the backfat thickness at farrowing and the litter size at weaning. Also, nothing is said about piglet mortality, we don’t know how many piglets were nursed until weaning, the differences in performance can be influenced by litter size, for example…

Lines 191-195: the sentence is not clear, please rephrase.

Figure 1 – I can’t see any scale bar

Line 323 – The reference is with a wrong parenthesis. Please check if 1.5% is really positive…

Some evolution/differences, for example, in enzymes of piglets could be addressed in the discussion, trying or explaining why some differences remain or disappear along time

Author Response

Reviewer #

1. Line 33 – are you sure about the word “adversely”?

RESPONSE: Thanks for your reminder. We have corrected it (line 33).

2. Line 47. The sentence started in this line is not clear probably because of missing words (after “important”?). Please rephrase.

RESPONSE: We have added (lines 47-48).

3. Line 80: Gilts are female pigs in their first gestation. Those were the animals used in the trial? If so, why do you say “average parity”? The average body weight should be given.

RESPONSE: Yes, your suggestion is appropriate, so we deleted average parity and added bodyweight and back fat thickness (line 80).

4. Line 83: replace “to the end of weaning” by “until weaning”

RESPONSE: I agree with your point, so I have corrected it (line 83).

5. Line 95: please explain better the criteria of cross-fostering, when you say “depending on their weight” your goal was?

RESPONSE: Cross-fostering was performed according to their availabitity and weight. There may be a difference in the litter size per gilt. The litter size determines the milk yield of gilts during lactation, so it is necessary to have the same number of litter per gilt by cross-fostering. In order to keep the similar litter weight and bodyweight as after cross-fostering, piglets with similar average litter weight were selected to be transferred in or out. The method of cross-fostering in this experiment mainly referred to previous studies (Lee et al., (2014), Journal of animal science & technology; Richert et al., (1997), Journal of animal science; Mateo et al., (2008), Journal of animal science). Most importantly, there was no significant difference in birth weight between the two treatment groups at the time of farrowing, also there was no significant difference in weight of piglets per litter after cross-fostering. Therefore, the method of cross-fostering in this experiment was reasonable and provided a rational study design to explain that piglet weight gain was due to milk yield and milk quality.

6. Lines 103-111: Several explanations are needed:

a) when you say “select average weight per litter” that means that it was a piglet with a body weight close to the average of that litter?

RESPONSE: We apologize for our written mistake. In fact, we selected close to the average weight of per treatment in per litter to collect samples. Now we have added relevant content (lines 106-111).

b) if so there were differences between selected piglets? (e.g. in a litter A the mean was 1.1kg while in litter B the mean was 1.4kg – this is related also with explanation in line 95);

RESPONSE: The selected piglets were differences in Con and Tau group, however, the bodyweight of piglets in same treatment group were close. In the original manuscript, we did not provide a clear description about the selection of piglets. Now we have added relevant content in line 106-111.

c)the blood was always obtained from the same piglets of each litter? If so couldn’t (shouldn’t) you perform repeated measures analyses of data?

RESPONSE: The blood obtained from slaughtered piglets of each litter at different times. So blood samples were collected from new born piglets (n = 8 per treatment, 1 piglet per litter which close to the average weight in per treatment), 7 D post-birth piglets (n = 8 per treatment, 1 piglet per litter which close to the average weight in per treatment) and 21 D post-birth piglets (n = 8 per treatment, 1 piglet per litter which close to the average weight in per treatment). We have added relevant content in line 106-111 and line 120-121.

d) the piglet blood samples collected on D1 were obtained before or after colostrum intake? If after, it could influence the enzymes results?

RESPONSE: I am very sorry that the details of our experiment have not been clearly explained. We have added “The new born piglets were removed from their mothers immediately after birth without sucking colostrum for blood sample collection.” in manuscript (line 111-112).

e) how were maintained the piglets after birth and before they could get access to colostrum?

RESPONSE: During the experiment, the piglets were placed in the incubator immediately after birth. After delivery finished (about 1-2 h), the piglets which close to the average weight in per treatment were selected to collect the blood samples and slaughter. Then all remaining piglets began to get access to colostrum. If some piglets could not get access to colostrum due to lack of vitality, we will take manual assistance.

f) the oxytocin administrated should be indicated in IU and the commercial product indicated.

RESPONSE: We have already corrected and added it (line 118).

7. Line 113: again clarify piglet selection. It was also important to indicate in the results section the weights (and variability) of slaughtered piglets at each age.

RESPONSE: Thank you very much for this comment. We have changed “At 1, 7, 21 D post-birth, selected an average weight of piglets slaughter in each group at each time point” to “Typical piglets with weight close to the average level of per treatment group were selected from each sow for slaughter at birth, 7 post-birth and 21 D post-birth (n = 8)” (lines 120-121).

8. Line 134-135: please rephrase or complete the sentence.

RESPONSE: Thank you for your suggestions. We have already revised (lines 142-143).

9. Line 181: please rephrase. Suggestion: on back fat thickness and weight, but the average daily feed intake was higher (p < 0.05) in the supplemented gilts during period lactation

RESPONSE: Thank you for your suggestions. We have already revised (lines 189-190).

10. Table 1: I consider important to have in this table: the backfat thickness at farrowing and the litter size at weaning. Also, nothing is said about piglet mortality, we don’t know how many piglets were nursed until weaning, the differences in performance can be influenced by litter size, for example…

RESPONSE: During the experiment, the mortality rate of piglets was very low between the two groups (In total 3 piglets died including 2 piglets in Con group and 1 in Tau group, respectively). The mainly reason cause of piglet death was crush ratio happened rather than nutrition, so we did not record the mortality rate. When piglet mortality happened, the dead piglets would be replaced by another piglet with similar bodyweight and age (At the beginning of the experiment, we chose 20 gilts located to Con and Tau groups, however, few gilts had lower litter size, so the date of each groups was collected from 8 gilts. The piglets which from the additional 4 gilts were used to cross-foster and alternate). We have added relevant content in line 96-97. Thanks for your suggestion. In addition, we have recorded weighted and back fat thickness (As shown in the table below). But those datas have been applied in another article and did not published yet. I'll just show you. We hope that these explanations are suitable there.

Items

Con

Tau

p-value

Gilt weight at day   75 of pregnancy, kg

189.06 ± 5.15

188.38 ± 2.33

0.905

Gilt weight after   farrowing, kg

195.00 ± 6.36

192.00 ± 3.09

0.744

Gilt weight at weaning, kg

186.44 ± 6.14

186.44 ± 2.97

1.000

Gilt back fat thickness at mating, mm

17.88 ± 0.58

18.63 ± 0.73

0.436

Gilt back fat thickness at farrowing, mm

18.75 ± 0.73

20.13 ± 0.81

0.227

Gilt back fat thickness at weaning, mm

16.13 ± 0.88

17.25 ± 0.86

0.375

11. Lines 191-195: the sentence is not clear, please rephrase.

RESPONSE: We are sorry about that we did not provide a clear description of results. We have re-described this part (lines 200-205).

12. Figure 1 – I can’t see any scale bar

RESPONSE: Thank you very much. We are sorry about that we did not provide a clear scale bar. I have been bolded the scale bar (line 219).

13. Line 323 – The reference is with a wrong parenthesis. Please check if 1.5% is really positive…

RESPONSE: I have corrected it to 0.3%. And 1.5% Tau to the diet of piglets resulted in significantly improved ileum villus height. Thank you (line 334).

14. Some evolution/differences, for example, in enzymes of piglets could be addressed in the discussion, trying or explaining why some differences remain or disappear along time

RESPONSE: Thank you for your suggestions. We have already added relative content in manuscript (lines 312-315).
